# Association of Daily Physical Activity and Sedentary Behaviour with Protein Intake Patterns in Older Adults: A Multi-Study Analysis across Five Countries

**DOI:** 10.3390/nu13082574

**Published:** 2021-07-27

**Authors:** Ilianna Lourida, Jolanda M. A. Boer, Ruth Teh, Ngaire Kerse, Nuno Mendonça, Anna Rolleston, Stefania Sette, Heli Tapanainen, Aida Turrini, Suvi M. Virtanen, Marjolein Visser, Carol Jagger

**Affiliations:** 1Population Health Sciences Institute, Newcastle University, Newcastle upon Tyne NE4 5PL, UK; 2National Institute for Public Health and the Environment (RIVM), PO Box 1, 3720 BA Bilthoven, The Netherlands; jolanda.boer@rivm.nl; 3Department of General Practice and Primary Health Care, School of Population Health, University of Auckland, Auckland 1023, New Zealand; r.teh@auckland.ac.nz (R.T.); n.kerse@auckland.ac.nz (N.K.); 4EpiDoC Unit, CEDOC, NOVA Medical School, Universidade Nova de Lisboa,1150-082 Lisbon, Portugal; nuno.mendonca@nms.unl.pt; 5Comprehensive Health Research Centre (CHRC), NOVA Medical School, Universidade Nova de Lisboa, 1169-056 Lisbon, Portugal; 6The Centre for Health, Tauranga 3110, New Zealand; a.rolleston@auckland.ac.nz; 7Council for Agricultural Research and Economics, National Research Centre for Food and Nutrition, Via Ardeatina 546, 00178 Rome, Italy; stefania.sette@crea.gov.it (S.S.); aida.turrini@gmail.com (A.T.); 8Finnish Institute for Health and Welfare (THL), FI-00271 Helsinki, Finland; heli.tapanainen@thl.fi (H.T.); suvi.virtanen@thl.fi (S.M.V.); 9Unit of Health Sciences, Faculty of Social Sciences, University of Tampere, FI-33014 Tampere, Finland; 10Centre for Child Health Research, University of Tampere and Tampere University Hospital, FI-33014 Tampere, Finland; 11The Science Centre of Pirkanmaa Hospital District, FI-33521 Tampere, Finland; 12Department of Health Sciences, Faculty of Science, The Amsterdam Public Health Research Institute, Vrije Universiteit Amsterdam, De Boelelaan 1085, 1081 HV Amsterdam, The Netherlands; m.visser@vu.nl

**Keywords:** protein intake, protein intake distribution, physical activity, sedentary behaviour, ageing, Newcastle 85+, DNFCS, FINDIET, INRAN-SCAI, LiLACS NZ

## Abstract

Physical activity and protein intake are associated with ageing-related outcomes, including loss of muscle strength and functional decline, so may contribute to strategies to improve healthy ageing. We investigated the cross-sectional associations between physical activity or sedentary behaviour and protein intake patterns in community-dwelling older adults across five countries. Self-reported physical activity and dietary intake data were obtained from two cohort studies (Newcastle 85+ Study, UK; LiLACS, New Zealand Māori and Non-Māori) and three national food consumption surveys (DNFCS, The Netherlands; FINDIET, Finland; INRAN-SCAI, Italy). Associations between physical activity and total protein intake, number of eating occasions providing protein, number of meals with specified protein thresholds, and protein intake distribution over the day (calculated as a coefficient of variance) were assessed by regression and repeated measures ANOVA models adjusting for covariates. Greater physical activity was associated with higher total protein intake and more eating occasions containing protein, although associations were mostly explained by higher energy intake. Comparable associations were observed for sedentary behaviour in older adults in Italy. Evidence for older people with higher physical activity or less sedentary behaviour achieving more meals with specified protein levels was mixed across the five countries. A skewed protein distribution was observed, with most protein consumed at midday and evening meals without significant differences between physical activity or sedentary behaviour levels. Findings from this multi-study analysis indicate there is little evidence that total protein and protein intake patterns, irrespective of energy intake, differ by physical activity or sedentary behaviour levels in older adults.

## 1. Introduction

The older population is growing rapidly and so does the challenge of keeping an increasing number of old and very old adults healthy. Older adults are at increased risk of poor health outcomes, including malnutrition, loss of muscle strength, and functional decline [1]. Diet and physical activity are modifiable lifestyle factors strongly related to health and physical function and they may contribute to strategies to improve healthy ageing. In particular, protein intake below the current recommended levels of 0.8 g/kg bodyweight/day (BW/d) has been associated with increased risk of adverse health outcomes, such as sarcopenia (age-related loss of muscle strength and mass), disability, frailty, and mortality in older adults [2,3,4,5,6,7]. Older adults usually eat less, including less protein, compared to their younger counterparts. This is largely due to physiological changes and medical conditions that may affect appetite and taste perception, reduced physical activity, and loss of independence, which may in turn limit shopping and food preparation [8]. Disease-related reduction in the utilization of available protein, and certain conditions, such as inflammatory diseases, may also increase protein requirements [9]. Adequate protein intake may play an important role in ageing-related outcomes and is important to preserve muscle mass, strength, and function.

Another area of interest besides protein quantity focuses on the distribution (timing) of protein intake. Several studies involving older adults have indicated that lunch and dinner protein intake were more likely to reach the recommended thresholds of ~30 g of high-quality protein required for muscle protein synthesis compared with breakfast [10]. It has been suggested that a diet pattern containing moderate amounts of protein (20–30 g) at each meal could be a more efficient strategy to optimise muscle protein synthesis compared to the same protein quantity consumed in a skewed pattern [11]. There is, however, mixed evidence for ‘pulse’ versus evenly distributed protein intake for improving lean muscle mass and strength [10,12].

Physical activity also stimulates muscle protein synthesis [8] and has been recognised as another modifiable factor associated with better health outcomes in all adults, including ageing-related outcomes, such as muscle strength decline, functional decline, frailty, and mortality [13,14,15]. The timing of protein intake has been suggested as a strategy to optimise the adaptive response to exercise, although evidence is mixed. Some studies, mainly in healthy younger adults, have shown that protein intake just before and/or immediately after training sessions or an even distribution of high-quality protein for 12 h post-exercise (compared to pulse intake) are associated with enhanced stimulation of muscle protein synthesis [16,17], whereas others have suggested that just the combination of resistance exercise and adequate total protein intake (but not protein timing) is the critical factor for muscle strength [18]. Furthermore, several studies have investigated the potential moderating effect of physical activity [2,5,6] or the synergistic effects of exercise and protein supplementation [19,20,21,22] on clinical outcomes (e.g., physical performance, disability, frailty) in older populations, again with mixed results. However, few studies have examined the association between physical activity and timing of protein intake itself in old (≥65 years) and very old (≥85 years) adults. Insight into the relationship between physical activity and protein intake behaviours in older adults can inform strategies to reduce malnutrition risk and improve the health status of the growing ageing population.

In this study, we investigated the association between physical activity or sedentary behaviour and both overall protein intake and the timing of intake in community-dwelling older adults from two cohort studies (≥85 years; United Kingdom and New Zealand), and three national food consumption surveys (≥65 years; the Netherlands, Finland, and Italy). More specifically, our objectives were to examine:

The relationship between physical activity and the likelihood of overall low protein intake, using the recommendation of <0.8 g/kg aBW/d as a definition of low protein intake.

The association between physical activity and the number of eating occasions providing protein (including snacks).

The association between physical activity and the likelihood of reaching the threshold of 20 or 30 g of protein in two or more meals. We also addressed whether older people with higher physical activity level more often reached the 20 g or 30 g threshold at breakfast.

The relationship between physical activity and the pattern of protein intake over the day, by examining whether older people with a higher physical activity level had a more skewed (pulse) protein intake pattern.

## 2. Materials and Methods

### 2.1. Study Populations

The main characteristics of the cohorts and surveys are provided below. The Newcastle 85+ Study is a longitudinal population-based study that approached all people turning 85 in 2006 (born in 1921) in Newcastle-upon-Tyne and North Tyneside (UK), for participation. The recruited cohort was socio-demographically representative of the general UK population at the time and did not include individuals with end-stage terminal illness. At baseline (2006–2007), multidimensional health assessment and complete general practice (GP) medical records data were available for 845 participants, of whom 722 community-living participants had complete dietary data, and body weight and height measurements [23,24].

The Life and Living to Advanced Age Cohort Study New Zealand (LiLACS NZ) recruited 937 octogenarians in 2010 who were: indigenous people of New Zealand (Māori, *n* = 421) born between 1920 and 1930, and non-indigenous (or non-Māori, *n* = 516) born in 1925, all living in the North Island of New Zealand. The study engaged multi-layer recruitment approaches to identify as many Māori and non-Māori people as possible. A comprehensive standardised questionnaire and physical assessments were completed face-to-face by trained interviewers at the participants’ local research clinic or residence annually. Medical history was ascertained from self-report, GP, and hospitalisation medical records. Dietary assessment was conducted at 12-month follow-up, with 578 participants (216 Māori, 362 non-Māori) completing both days of the dietary assessments. In total, 536 participants (183 Māori, 353 non-Māori) had complete diet and physical activity data. Details of the recruitment strategies and the cohort profile for both sample groups have been reported [25,26].

The Dutch National Food Consumption Survey (DNFCS)-Older Adults 2010–2012 is a nationwide cross-sectional study investigating the diet of community-dwelling adults aged 70 years and older in the Netherlands. Data were collected from October 2010 to February 2012 in 15 municipalities of 5 different regions in the Netherlands. Of the 2848 older adults eligible to participate, 739 participants were included in the DNFCS-Older Adults 2010–2012. Detailed information on the study design and data collection has been described previously [27].

The Finnish National FINDIET surveys are part of the national FINRISK study, a five-yearly cross-sectional population survey assessing risk factors of chronic diseases. For the FINRISK survey, a random sample of persons aged 25–74 years, stratified by sex, area, and 10-year age groups, was drawn from the population register. The survey covers five study areas in Finland representing 35% of the population. The survey includes a health examination at the local health centre, and participants are asked to complete a questionnaire that covers socio-economic factors, medical history, perceived health, and lifestyle [28,29]. From the FINRISK sample, 33% were invited to the FINDIET Survey. For the current analysis, we pooled the data of participants aged 65 years and over (*n* = 876) from the 2007 and 2012 surveys.

The third Italian National Food Consumption Survey, INRAN-SCAI, was conducted between October 2005 and December 2006 on 1300 randomly selected households stratified into the four main geographical areas of Italy (north-west, north-east, centre, south and islands). Of the 3323 individuals participating in the survey [30], 518 participants aged 65 years and over had complete dietary data and were included in this analysis.

### 2.2. Dietary Intake

Information on dietary intake was obtained from 24 h recalls or food diaries as described below.

Newcastle 85+ Study: Dietary intake was assessed according to 24 h multiple pass recall on 2 non-consecutive weekdays at least one week apart. The two recalls were conducted by trained research nurses, with portion sizes estimated by a photographic food atlas. Energy and protein intake were estimated using McCance and Widdowson’s sixth edition food composition tables [31].

LiLACS NZ: Dietary intake was assessed using the same method as the Newcastle 85+ Study, by 24 h multiple pass recall on 2 different days, with coding of foods by nutritionists experienced in dietary data coding. FOODfiles (2010), an electronic subset of data from the New Zealand Food Composition Database (NZFCDB), was used as the main source of food composition data [32].

DNFCS: Trained dieticians measured dietary intake during home visits by means of two face-to-face non-consecutive dietary record assisted 24 h recalls. The two 24 h dietary recalls took place within a period of two to six weeks, with a mean interval of four weeks. Consumption on Sunday to Friday was recalled the next day, while consumption on Saturday was recalled on the following Monday. On the day to be recalled, participants filled in a food diary, which was used as a memory aid during the 24 h recall and as a check for the use of household measures. Intakes of energy and nutrients were calculated using an extended version of the Dutch Food Composition Table of 2011 [27].

FINDIET: A 48 h dietary recall interview was conducted face-to-face by trained nutritionists. A validated picture booklet and household measures were used for portion size estimation. All interviews were carried out between January and April. The Finnish national food composition database was used for the coding and calculation of nutrient intake [28].

INRAN-SCAI: Food consumption data were collected at the individual level for three consecutive days using the estimated food record method with a semi-structured diary. All foods and drinks consumed (including tap and bottled water), both at and outside home, were recorded by each participant using household measures and portion sizes estimated according to detailed guidance notes and a photograph atlas. Individual food records were converted into energy and nutrient intakes with the use of updated national food composition databases [33,34].

Energy intake was expressed in kilocalories (kcal) across studies.

### 2.3. Protein Intake

In addition to calculating mean protein intake in grams per day (or grams per adjusted body weight per day), we defined “low” protein intake as below 0.8 g of protein per kilogram of adjusted body weight per day, the current European Recommended Dietary Allowance (RDA) as indicated by the European Food Safety Authority [35]. For participants with a body mass index (BMI) outside the healthy range of 18.5–24.9 kg/m^2^ for adults aged ≤ 70 years and 22.0–27.0 kg/m^2^ for adults > 70 years, we replaced actual body weight by adjusted body weight (aBW), this being the nearest body weight that would place the participant with an undesirable body weight in the healthy BMI range [36].

Each moment where foods or drinks were ingested at the same day, time, and place was considered as one eating occasion, and those providing protein were summed per person and per day to obtain the number of eating occasions providing protein. In addition, in all included studies, the amount and proportion of protein intake was calculated for six predefined eating occasions (breakfast, morning snack, lunch, afternoon, dinner, evening snack), with an additional eating occasion of an overnight snack in the Newcastle 85+ Study. Further details are provided in the Appendix A.

In the Newcastle 85+ Study and LiLACS NZ, we calculated the number of main meals (breakfast, lunch, dinner) that met protein thresholds of 20 and 30 g (categorised as 0, 1, 2+ meals), as well as the number of participants who had at least one of those eating occasions. In the national surveys, we determined for each meal occasion whether the meal provided more than the recommended thresholds of 20 g or 30 g of protein, as well as the number of main meals providing more than the recommended thresholds.

Skewness of protein intake over the day was determined using the coefficient of variation (CV). A higher CV can be interpreted as more variation in intake across the eating occasions (more skewed) and therefore pulse eating. In all studies, the CV was calculated for all eating occasions and for main meals only.

Finally, in the national surveys, protein intake patterns were also defined by the timing of protein intake, i.e., by calculating the amount (grams) and proportion (%) of protein ingested during every hour of the day (not possible in Newcastle 85+ Study and LiLACS NZ). The proportion was calculated as the percentage of total protein intake of that day. In DNFCS, time-of-day (per full hour) was reported for each occasion food was consumed, with ‘hour = 07:00′ indicating that the occasion started between 06:30 and 07:29 h. Data of INRAN-SCAI and FINDIET were recoded to get similar hourly intervals. Hours providing <1% of total protein intake were excluded from the analyses.

### 2.4. Physical Activity

Physical activity was self-reported in all the studies, though harmonisation across studies was not possible due to the differences in physical activity items. Each study therefore used its own definition.

Newcastle 85+ Study: Self-reported physical activity was assessed with a purpose-designed physical activity questionnaire across four waves measuring the frequency and intensity of physical activity conducted in a week. At the third wave (age 88), the self-reported physical activity measure was collected alongside accelerometry data, and was found to be strongly associated with accelerometry measures, including the daily sedentary time, low-intensity physical activity, activities of daily living, and walking [37]. In the current analyses, we calculated tertiles of the self-reported physical activity scores at baseline and categorized participants as low (scores 0–2), intermediate (scores 3–7), and high physical activity (scores 8–18). To get a score of 8–18, an individual would have to do a minimum of moderately energetic activities ≥3 times a week (if they did vigorous activities hardly ever or never). To have a low score (0–2), the most they would do is moderately energetic once, twice, or three times a month. Physical activity data were missing for one participant, thus the sample size for the current analysis was 721.

LiLACS NZ: Physical activity was assessed with the Physical Activity Scale for the Elderly (PASE) [38,39]. PASE consists of 10 items used to identify household-, occupational-, and leisure-related activity, and the duration of each activity over a one-week period. For the current analysis, we categorized participants into low (scores <53), intermediate (scores 53–107), and high (scores >107) physical activity groups according to ethnic-specific tertiles of the PASE. Typically, low physical activity involved outdoor gardening, leisure walking 5–7 days a week for 1–2 h per session, and light sport (e.g., bowl) 1–2 days a week for 1–2 h. A high physical activity score would involve additional lawn or yard care the past week, and strength/endurance exercises 3–4 days a week for 1–2 h per session.

DNFCS: During the first home visit, an interviewer administered the Short QUestionnaire to ASsess Health enhancing Physical Activity (SQUASH) for adults [40], in which the final question asked how many days per week respondents were doing at least 30 min of moderately intense physical activity, both in summer and during the rest of the year. The physical activity level of participants was based on the average number of days with at least 30 min of moderately intense physical activity, and classified into: low (inactive: 0 days), intermediate (semi-active: 0.5–4.5 days), or high (norm-active: 5.0 or more days). Data on physical activity were missing for one participant, resulting in an analytic sample of 738.

FINDIET: Leisure time physical activity was self-reported with four response options: low (mostly inactive, e.g., reading and watching television), intermediate (at least 4 h of exercise weekly, e.g., walking, cycling, fishing, hunting, or light gardening), high (e.g., running, skiing, swimming, or more vigorous sports more than 3 h per week), and very high (competitive exercising almost daily). For the analysis, we combined the high and very high categories into high physical activity. Nine participants had missing data on physical activity, thus the sample size for the current analysis was 867.

INRAN-SCAI: Self-reported physical activity was determined by hours per day of light physical activity, and categorised into low (no/light physical activity), intermediate (>0 and <2 h/day), and high (≥2 h day). Eight participants had missing values on physical activity (analytic sample size = 510).

### 2.5. Sedentary Behaviour

Information on sedentary behaviour was only available in FINDIET and INRAN-SCAI.

In INRAN-SCAI, hours of sedentary behaviour per day was self-reported according to three response categories: <4 h/day, 4–6 h/day, and >6 h/day. Self-reported sedentary behaviour was determined by five questions in FINDIET asking how many hours respondents sit on average on a weekday during the workday in office or equivalent, at home watching television or videos, at home at a computer, in a vehicle, and elsewhere. Reported hours were summed and categorized into the same three categories as those in INRAN-SCAI. Information on sedentary behaviour was missing for 25 participants of INRAN-SCAI and 78 participants of FINDIET, resulting in analytical sample sizes of 493 and 798, respectively.

### 2.6. Socioeconomic, Lifestyle, and Health Factors

Socioeconomic factors included self-reported education (all studies; years spent in education or low/intermediate/high), deprivation (Newcastle 85+ Study: Index of Multiple Deprivation, poor/intermediate/affluent areas; LiLACS NZ: NZ Deprivation Index tertiles, high/mid/low deprivation), household income (DNFCS, FINDIET; low/high/unknown), and living arrangements (all studies; living alone/not living alone or living together/not living together). Lifestyle factors included smoking (all studies; yes/no or never/former/current smoker) and alcohol (Newcastle 85+ Study, LiLACS NZ; yes/no). Further details are provided in the Appendix A.

BMI was calculated as weight in kg/height in m squared. BMI was categorised into the following categories for adults >70 years: underweight (<22.0 kg/m^2^), normal weight (22.0–26.9 kg/m^2^), overweight (≥27.0 kg/m^2^), and additionally in the Newcastle 85+ Study and LiLACS NZ obese (≥30 kg/m^2^) [36]. In the national surveys, the BMI categories for adults ≤ 70 years were: underweight (<18.5 kg/m^2^), normal weight (18.5–24.9 kg/m^2^), and overweight (≥25.0 kg/m^2^).

Physical and oral health factors were only available in the Newcastle 85+ Study and LiLACS NZ and included a disease count (<2, 2, or ≥3 diseases) created by scoring seven common chronic diseases as present or absent (cardiac, respiratory, and cerebrovascular disease, hypertension, arthritis, diabetes mellitus, and cancer in the past 5 years), a disability score (none, 1–6, 7–12, or 13+ activities) calculated from the total of 17 activities of daily living performed with difficulty or requiring an aid or appliance or personal help (Newcastle 85+ Study), the number of activities in the Nottingham Extended Activities of Living Scale [41] performed with assistance (LiLACS NZ), and swallowing problems (yes/no) from questions ascertaining dry mouth and difficulty swallowing for other reasons (Newcastle 85+ Study), and coughing/choking/pain when swallowing food or fluids (LiLACS NZ).

### 2.7. Statistical Analysis

Given the differences between the studies, especially in terms of the age ranges, as well as the consistency with previous publications involving protein intake, we analysed the national surveys and the Newcastle 85+ and LiLACS NZ cohort studies separately. Analytic methods are therefore specific to each dataset and are detailed individually below.

In the Newcastle 85+ Study and LiLACS NZ, individuals were the units of observation in the analyses (where protein was calculated as the average intake over the two recall days). Participant characteristics are shown as means and standard errors for continuous variables, and as percentages for categorical variables (with the number of participants in parentheses) across physical activity tertiles. After testing for normality using the Shapiro–Wilk test, we assessed differences between physical activity tertiles with the Kruskal–Wallis test for continuous non-normally distributed variables and the chi-squared test (χ²) for categorical variables.

Analyses were conducted through a series of logistic regression models with the dependent variable as protein intake status (<0.8 g/kg aBW/day versus ≥0.8 g/kg aBW/day); number of eating occasions that include protein categorized as 0–4, 5, 6–7 eating occasions (ordered logistic regression); number of main meals (breakfast, lunch, dinner) where the 20 or 30 g threshold is reached, classifying the total as 0, 1, 2+ (ordered logistic regression); and whether the 20 g or 30 g threshold is reached at breakfast (yes/no). All models were adjusted for (1) sex, (2) confounding factors (education, living arrangement, deprivation, smoking, alcohol), (3) energy intake (kcal), and (4) additional health factors (disease count, disability, swallowing problems). For the final objective, we characterised the protein intake distribution using the coefficient of variation (CV; standard deviation/total protein intake) between all eating occasions and for main meals only, both in quartiles and fitted ordered logistic regression models. Given the lower physical activity levels in very old women compared to men, we conducted sensitivity analyses by refitting all models using sex-specific tertiles of physical activity. Statistical analyses were conducted using Stata v.15 [42] and IBM SPSS statistics, version 27, and *p* < 0.05 was considered statistically significant.

In the national dietary surveys, participant characteristics were expressed as frequency and percentage for categorical data, and mean and standard deviation for continuous data. For results regarding dietary intake, we used recall days as the units of observation, to allow for differences in protein intake patterns between days, and to minimise any attenuation of the effect of physical activity and sedentary behaviour on protein intake patterns from averaging protein intake patterns over recall days. Therefore, the reported sample sizes in tables refer to the number of recall days per survey for participants with non-missing physical activity data, i.e., 2 days *×* 738 participants in DNFCS, 2 days × 867 participants in FINDIET, 3 days × 510 participants in INRAN-SCAI. We compared protein intake by physical activity level (and sedentary behaviour level for FINDIET and INRAN-SCAI) by repeated-measures ANOVA, with an unstructured covariance matrix to account for within-person correlation, and adjusting for (1) age and sex, (2) confounding factors (marital status, education, smoking, and household income (where available)), and (3) energy intake. Two sensitivity analyses were undertaken, one excluding special days, and another excluding days on which special diets were consumed (which was not possible for FINDIET). All analyses were performed using SAS software^®^ (Version 9.4 of the SAS System for Windows).

## 3. Results

### 3.1. Participant Characteristics

Participant characteristics in the five studies are presented in Appendix A. Participants from the Newcastle 85+ Study in the low physical activity tertile were less likely to be men or to drink alcohol, have a lower adjusted body weight, and have a greater number of chronic conditions, disabilities, and swallowing problems. In LiLACS NZ, similar to the Newcastle 85+ Study, a higher physical activity level was observed in men compared to women (non-Māori only), and an inverse relationship between physical activity and the number of conditions and the number of disabilities (Māori and non-Māori), and a positive association between alcohol consumption and physical activity (non-Māori only) (Appendix A). For both cohort studies, there was no evidence of differences in years of education, living arrangements, deprivation index, smoking status, or BMI by physical activity tertile (Appendix A). In all three national surveys, participants with a low physical activity level were significantly older and had a significantly higher BMI than participants with an intermediate and/or high physical activity level (Appendix A). Marital status did not differ according to physical activity level. For the other characteristics, associations were less consistent between countries, possibly because of the different definitions used.

In both the Finnish and Italian survey, there was no difference in mean age between the three groups with various levels of sedentary behaviour (Appendix A). In the Italian, but not in the Finnish, survey, participants who were the most sedentary (six or more hours per day) were significantly younger and a larger proportion were married or living together than participants with less than four hours of sedentary behaviour per day. In the Finnish survey, the most sedentary older adults were more likely to be male, higher educated, and had a high income. There were also more former and current smokers among the most sedentary Finnish older adults, and there was a non-significant trend for this in Italian older adults.

### 3.2. Physical Activity and Protein Intake

Energy and protein intakes were higher in Newcastle 85+ Study and in LiLACS NZ (non-Māori and Māori) participants in the high physical activity tertile, with significant differences between tertiles in all intakes except for the grams of protein consumed at breakfast (both studies) and in protein expressed as g/kg aBW/day (LiLACS NZ Māori) (Appendix A). In adjusted analyses, there was a significant positive relationship between mean protein intake expressed as grams/day or g/kg aBW/day and physical activity in most of the studies (Newcastle 85+ Study, LiLACS NZ Māori, DNFCS, FINDIET) (Table 1). In the sensitivity analyses with sex-specific tertiles of physical activity, there was no longer a significant positive relationship between protein intake in grams/day and physical activity in LiLACS NZ Māori participants. Additionally, in the national surveys, the association disappeared after further adjustment for energy intake (Appendix A), and conclusions remained after additionally excluding special days or participants on a special diet in the Dutch survey (not possible in FINDIET) (Appendix A).

The prevalence of low protein intake (<0.8 g/kg aBW/d) was slightly higher in both the Newcastle 85+ Study and LiLACS NZ participants in the low physical activity tertile compared to those in the intermediate and high physical activity tertiles (Appendix A). However, differences were not statistically significant and remained so in models adjusting for a number of confounding factors, and energy intake (Appendix A). The same pattern was observed in sensitivity analyses with sex-specific tertiles of physical activity in the Newcastle 85+ Study and LiLACS NZ. Similarly, in the Italian survey INRAN-SCAI, no significant differences between physical activity levels were observed in the percentage of days on which protein intake was below the recommendation of 0.8 g/kg aBW/d. In contrast, participants with a low physical activity level in DNFS and FINDIET had more days where protein intake was below 0.8 g/kg aBW (Appendix A). These differences also became non-significant after adjustment for energy intake and additional exclusion of special days and days on which a diet was followed.

In the FINDIET and INRAN-SCAI, both protein intake and the percentage of days with a low protein intake did not differ according to level of sedentary behaviour (Appendix A).

### 3.3. Physical Activity and Number of Eating Occasions Providing Protein

Evidence of a relationship between physical activity and the number of eating occasions providing protein was mixed between the studies. Older adults in the high physical activity level were significantly more likely to have a higher number of eating occasions containing protein compared with participants in the intermediate or low levels in the Newcastle 85+ Study, LiLACS NZ Non-Māori, the FINDIET and INRAN-SCAI (Appendix A). However, after further adjustment for energy intake (and exclusion of special days or diet days in national surveys), the association remained only for LiLACS NZ non-Māori, Italian, and Finnish older adults (Table 2 and Table 3). For LiLACS NZ Māori, and Dutch older adults, no differences were observed for the number of protein-providing eating occasions. Conclusions remained the same in the Newcastle 85+ Study and LiLACS NZ when sex-specific tertiles of physical activity were examined.

Among Finnish older adults, no differences according to sedentary behaviour were observed in the number of eating occasions providing protein (Appendix A). Italian older adults who were sedentary for more than 6 h per day had, however, fewer eating occasions providing protein (3.7 ± 0.1) than older adults with 4–6 h (4.1 ± 0.1) or less than 4 h of sedentary behaviour (4.2 ± 0.1).

### 3.4. Physical Activity and Reaching Protein Thresholds

Only in the Netherlands (DNFCS) and LiLACS NZ Māori participants was there any evidence of an association between physical activity level and the percentage of days on which at least two eating occasions or main meals provided at least 20 g of protein (Newcastle 85+ Study, LiLACS NZ: Appendix A; National surveys: Table 4). For LiLACS NZ Māori participants, the association was no longer evident after adjustment for potential confounders. However, in the Netherlands (DNFCS), the differences were more pronounced when looking at main meals only, and differences remained significant after adjustment for energy intake (Appendix A). However, there was no evidence of an association when special or diet days were excluded (Appendix A) or when the protein threshold was raised to 30 g (National surveys: Table 4), or for protein intake thresholds at breakfast (data not shown). Conclusions remained the same from the sensitivity analyses results in the Newcastle 85+ Study and LiLACS NZ using sex-specific tertiles of physical activity across models.

For sedentary behaviour, evidence of an association was only observed in Italy (INRAN-SCAI). The most sedentary Italian older adults (>6 h per day) were more likely to have two or more eating occasions or main meals providing more than 20 or 30 g of protein (Appendix A). Conclusions remained after adjustment for energy intake, but there was no evidence of an association after the exclusion of special days, or for protein intake thresholds at breakfast (data not shown).

### 3.5. Physical Activity and Pattern of Protein Intake over the Day

Whether the distribution of protein intake was more or less skewed by physical activity level was assessed by calculating the coefficient of variation (CV) of protein intake across all meals and then for main meals only. Only in Italy was there evidence of an association between a higher physical activity level or sedentary behaviour and the pattern of protein after adjustment for confounders (Figure 1 and Appendix A). However, it was notable that older adults in New Zealand and the Netherlands appeared to have a more even distribution of protein (lower CV) in their main meals than older adults in the UK, Finland, or Italy (Figure 1). In Italy, older people with a high physical activity level had a significantly lower CV, suggesting a more even distribution of protein intake over the meal occasions, probably as a result of the somewhat higher protein intake at breakfast in the most physically active (Figure 2). Older adults with a high level of sedentary behaviour had a significantly higher CV over all eating occasions, suggesting a less even distribution of protein intake across meals (Appendix A). These conclusions remained after additional adjustments and exclusions, and appeared due to the difference in protein intake for morning snacks (Appendix A).

In the sensitivity analyses, there were no significant associations between sex-specific tertiles of physical activity and the distribution of protein intake (Newcastle 85+ Study and LiLACS NZ). The only exception was a marginal association for Māori participants (LiLACS NZ) in the low physical activity level, suggesting a more even distribution of protein intake in their main meals (data not shown).

Although there were differences in the mean amount of protein ingested (Figure 2) and in the proportion of protein ingested (Appendix A) for several hours of the day in each country, differences by physical activity level were small and findings were inconsistent across countries.

Similar small and inconsistent findings were observed when looking at differences in dietary patterns over hours of the day according to sedentary behaviour (data not shown), suggesting that protein intake patterns over the day do not differ substantially among older adults who differ in their physical activity levels or sedentary behaviour.

## 4. Discussion

In this multi-study analysis, we examined the associations of physical activity and sedentary behaviour with protein intake patterns in community-dwelling older adults across five countries, including a substantial number of very old adults. Results from two cohort studies and three dietary surveys showed that there appears to be a positive relationship between physical activity and levels of protein intake in older people across the different countries. However, this association seems to be mostly explained by a higher energy intake (greater food consumption) in the most physically active. In contrast, the time spent on sedentary behaviour was not consistently associated with protein intake. These findings importantly suggest that physically active older adults will be more likely to meet the protein requirements compared to physically inactive older adults, likely due to their higher energy intake. Physical activity may therefore not only have a direct positive impact on relevant clinical outcomes due to its anabolic effects on increasing muscle mass and muscle strength [8], but also indirectly through a higher energy and protein intake.

Older adults in the UK, Italy, and Finland who were physically active appeared to have more eating occasions that contain protein than older adults who were less physically active. This positive relationship between the number of eating occasions containing protein and levels of physical activity again appeared to be accounted for by greater energy intake in the more physically active (and other confounding factors in some of the study populations). These results suggest that physically active older adults eat more often and are thereby more likely to have a higher number of eating occasions containing protein. This is consistent with studies supporting increased energy requirements in healthy older adults following a high-intensity resistance training programme [43], although the picture for old or very old adults with varying number of medical conditions and high levels of habitual physical activity may be different. While we used the current RDA of 0.8 g/kg BW/day, it has been suggested that older adults have greater protein needs than the current recommendations. The PROT-AGE study group has recommended an increased protein intake to achieve at least 1.0–1.2g protein/kg BW/day for active older adults, whereas for most older adults with a chronic disease, the protein intake recommendation is even higher (i.e., 1.2–1.5 g/kg BW/day) [44]. In our analysis, only Māori participants (LiLACS NZ) in the high activity group achieved those levels (1.3 g/kg aBW/day, Table 1). As an increasing amount of evidence suggests that older individuals require a higher protein intake to maintain muscle mass and prevent chronic diseases, the use of higher protein cut-offs may also be more appropriate to improve our current understanding of the relationship between protein intake, activity, and healthy ageing.

Despite this, our results do not suggest that physically active older adults are also more likely to achieve more meals that contain an amount of protein reported to optimally stimulate muscle protein synthesis [45]. This relationship was evident only for older people in the Netherlands (and only for the 20 g of protein threshold) and for the most sedentary older adults in Italy. These rather conflicting findings between activity levels indicate that other factors not accounted for in our analyses, such as personal, social, cultural, or other dietary pattern differences (e.g., animal- or plant-based protein sources), may also play a role in protein intake and activity. More studies are needed to clarify these findings.

Recommendations for an evenly distributed protein intake state that an intake of 25–30 g per meal has potential benefits on muscle protein synthesis, lean body mass, and frailty [11,12,46]. In our analysis, there appeared to be a somewhat skewed distribution of protein intake across the day in most countries (except Finland), with the majority of protein consumed at the midday and/or evening meals. However, there was little evidence that these patterns differed depending on the physical activity level or sedentary behaviour of participants. Additional studies investigating the timing and intensity of physical activity in relation to protein intake patterns may shed more light on this association and the potential benefits on physical function of older adults through targeted lifestyle and nutritional interventions.

Several randomised controlled trials have examined the effectiveness of interventions combining protein supplementation and exercise (vs. exercise only) on physical function outcomes with mixed findings. Some studies have shown significant improvements in muscle strength and gait speed (important measures of survival in older adults) [19,21,47], whereas others found no evidence of significant differences between intervention groups in physical performance measures, including grip strength and various walking tests [20,22,48,49]. Overall, further studies addressing low power to detect associations, compliance rates, and longer follow-up periods are needed to clarify whether approaches combining protein-based supplements and exercise programs are viable strategies to improve and maintain physical function in older adults. A better understanding of the associations between the physical activity/sedentary behaviour and protein intake patterns of older adults could provide valuable additional information for the design of future interventions aiming to reduce functional decline and promote healthy ageing in the older population.

To our knowledge, this analysis is the first attempt of using data from five different countries to investigate physical activity and protein intake patterns in community-dwelling older adults covering a range of demographic, socioeconomic, cultural, health, diet, and physical activity characteristics. The studies relied on self-reported measures of dietary intake and although commonly used in this type of research, they are prone to recall bias (especially in older adults) and social desirability bias. Although we tried to harmonise variables and analytic methods across the studies, this was not always possible. For example, in the Newcastle 85+ Study and LiLACS NZ, protein intake was analysed as the mean of 2-day intakes with the number of individuals as the unit for analysis, whereas in the national surveys, the unit for analysis was the number of recall days per survey. It is therefore possible that associations in the Newcastle 85+ Study and LiLACS NZ may be attenuated. However, the different dietary assessment methods used in the studies within this analysis are comparable and they allowed calculation of the protein intake in the same way, including protein distribution over the day (protein CV), which are strengths of this analysis. That said, it should be noted that an intake below the RDA does not necessarily mean that protein intake is inadequate (compared to the needs). Protein inadequacy on an individual level could not be assessed from the available data. Additionally, neither the cohort nor the dietary surveys were able to objectively measure physical activity by means of accelerometers (on the same day as the dietary assessment), and each study used its own definition of physical activity levels. Nevertheless, the Newcastle 85+ Study did have accelerometry data and self-reported physical activity (but not dietary data) at the 36-month interview, which provided a validation of the self-reported physical activity measure [37]. Although most of the studies used items from validated physical activity questionnaires that provide a good estimate of participant activity levels, they remain subject to self-report bias and may deviate from actual behaviour. Future studies with sufficient sample sizes that include measurements of physical activity using accelerometers paralleled with 24 h information on dietary intake on multiple days are needed to gain further insight in the interrelationship between daily physical activity and dietary patterns. Linked to the above, the cross-sectional analysis limits the ability to assess the temporality of the association and, as already mentioned, the potential importance of timing of protein intake in relation to a bout of physical activity. Since men are more likely to engage in higher activity levels than women, we included sensitivity analyses examining associations between sex-specific tertiles of physical activity and protein intake (in two of the studies). However, there were no substantial differences to the presented results based on the combined sample physical activity tertiles. Sex differences in dietary behaviour and intake also exist. We did not stratify our analyses by sex due to insufficient power in the cohort studies, but we acknowledge this as a limitation that should be addressed in future studies. All study samples included in our analysis are population-based, and therefore the results should be relevant to the respective populations. Few studies include octogenarians and thus this analysis represents a significant step forward in understanding protein intake and activity in the very old.

In summary, participants across studies had on average sufficient protein intake according to recommendations and a range of physical activity levels. More physically active participants were more likely to have more eating occasions containing protein and higher total protein intake, but these associations were mostly explained by higher energy intake. Evidence for older adults with higher physical activity or less sedentary behaviour achieving more meals containing adequate levels of protein is mixed. We observed an uneven protein distribution across the day, confirming previous research showing that the majority of protein intake occurs at lunch and evening meals, with little evidence that these patterns differ by physical activity or sedentary behaviour level. Overall, our findings indicate that any advice regarding protein intake could likely be similar for inactive and active older adults. Given the high public interest of strategies to delay the decline of physical function and disability in older adults and their relationship to protein intake, further research into the potential effect of the timing and type of physical activity in relation to protein intake patterns and health outcomes is important to clarify associations and to inform ongoing efforts to design interventions aiming to maintain or improve the health status of the older population.

## Figures and Tables

**Figure 1 nutrients-13-02574-f001:**
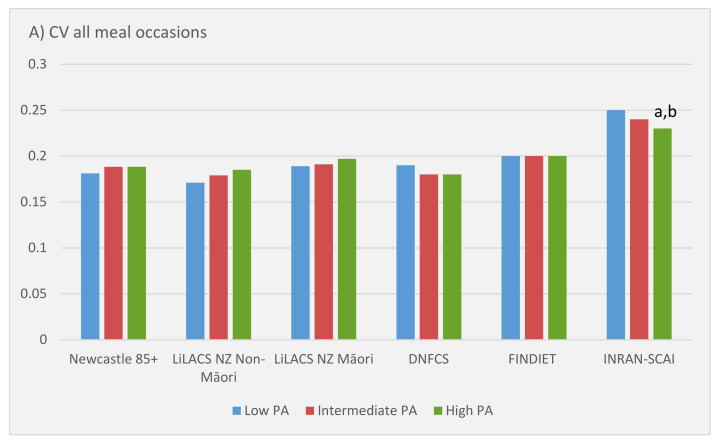
Coefficient of variation of protein intake over (**A**) all eating occasions or (**B**) main meals only according to physical activity in the Newcastle 85+ Study, LiLACS NZ, Netherlands (DNFCS), Finland (FINDIET), and Italy (INRAN-SCAI). A lower CV indicates a more evenly distributed protein intake. Newcastle 85+ Study and LiLACS NZ adjusted for age (LiLACS only), sex, educational level, deprivation, living alone, smoking, and alcohol. DNFCS, FINDIET, INRAN-SCAI adjusted for age, sex, educational level, marital status, and smoking. ^a^ Differs from those with a low physical activity level *p* < 0.05.^b^ Differs from those with an intermediate physical activity level *p* < 0.05.

**Figure 2 nutrients-13-02574-f002:**
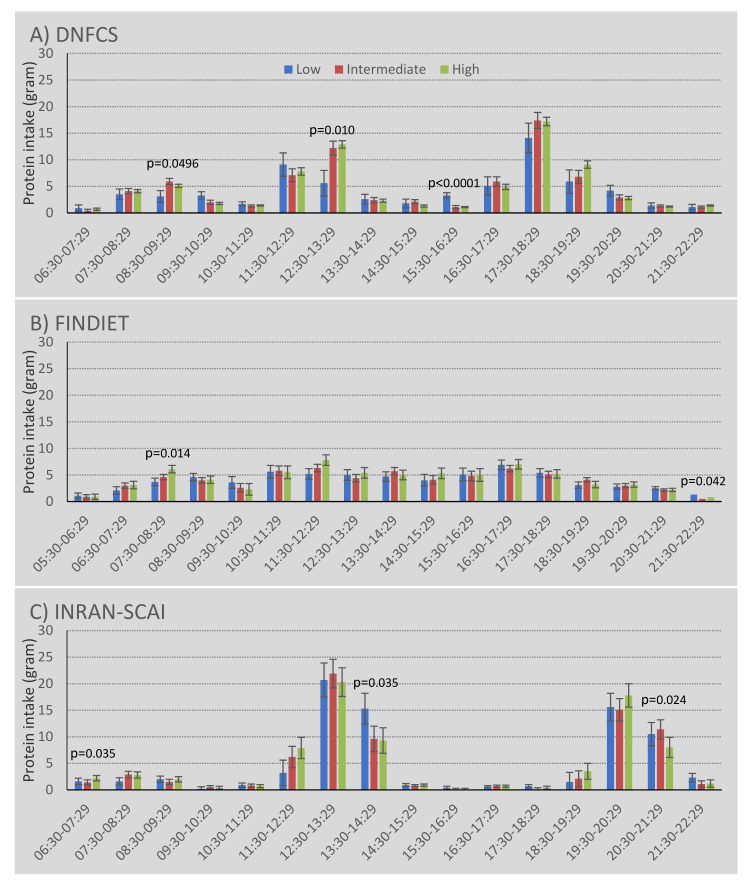
Amount of protein intake across time of the day and stratified by physical activity level among community-dwelling older adults in (**A**) the Netherlands (DNFCS), (**B**) Finland (FINDIET), and (**C**) Italy (INRAN-SCAI). Adjusted for age, sex, educational level, marital status, (household income), and smoking.

**Table 1 nutrients-13-02574-t001:** Protein intake according to physical activity in community-dwelling older adults from the UK (Newcastle 85+ Study), New Zealand (LiLACS NZ), Netherlands (DNFCS), Finland (FINDIET), and Italy (INRAN-SCAI).

	Physical Activity Level			
	Low	Intermediate	High	*p*-Value
Newcastle 85+ Study * (*n* = 721)	*n* = 184	*n* = 335	*n* = 202	
Protein intake—grams/day	59.4 ± 1.6	63.8 ± 1.1	68.7 ± 1.4	0.0001
Protein intake—g/kg/aBW/day	0.94 ± 0.03	0.99 ± 0.02	1.07 ± 0.02	0.0003
LiLACS NZ Non-Māori * (*n* = 353)	*n* = 123	*n* = 120	*n* = 110	
Protein intake—grams/day	56.6 ± 2.1	56.4 ± 2.0	60.8 ± 2.0	0.212
Protein intake—g/kg/aBW/day	1.1 ± 0.03	1.0 ± 0.03	1.1 ± 0.03	0.181
LiLACS NZ Māori * (*n* = 183)	*n* = 58	*n* = 57	*n* = 68	
Protein intake—grams/day	52.4 ± 4.8	51.1 ± 4.6	66.0 ± 4.1	0.031
Protein intake—g/kg/aBW/day	1.1 ± 0.10	1.1 ± 0.09	1.3 ± 0.08	0.334
DNFCS ^†^ (*n* = 1476)	*n* = 64	*n* = 252	*n* = 1160	
Protein intake—grams/day	68.7 ± 3.3	75.1 ± 1.8	76.5 ± 1.0	0.055
Protein intake—g/kg/aBW/day	0.94 ± 0.05	1.05 ± 0.03	1.07 ± 0.01	0.044
FINDIET ^†^ (*n* = 1734)	*n* = 286	*n* = 1088	*n* = 360	
Protein intake—grams/day	67.5 ± 2.4	68.2± 1.7	72.5 ± 2.3	0.085
Protein intake—g/kg/aBW/day	0.97 ± 0.03	0.99 ± 0.02	1.06 ± 0.03	0.039
INRAN-SCAI ^†^ (*n* = 1530)	*n* = 201	*n* = 642	*n* = 687	
Protein intake—grams/day	79.1 ± 3.3	77.8 ± 2.7	79.3 ± 2.8	0.73
Protein intake—g/kg/aBW/day	1.16 ± 0.05	1.15 ± 0.04	1.18 ± 0.04	0.70

Presented as mean ± standard error. * Adjusted for sex, living arrangement, education, deprivation, smoking and alcohol; differences between physical activity tertiles assessed by the Kruskal–Wallis test; *n* is number of participants with non-missing physical activity data; ^†^ Adjusted for age, sex, educational level, marital status, household income, and smoking; *n* is number of recall days per survey for participants with non-missing physical activity data; differences between physical activity tertiles assessed by repeated-measures ANOVA, with an unstructured covariance matrix to account for within-person correlation; *p* < 0.05 indicates there are significant differences between physical activity groups.

**Table 2 nutrients-13-02574-t002:** Odds ratios (ORs) and 95% CI for the association between physical activity tertiles and number of eating occasions containing protein in the Newcastle 85+ Study and LiLACS NZ.

	Physical Activity Level		
	Low	Intermediate	High
Newcastle 85+ Study			
OR (95% CI)			
Model 1	0.86 (0.62–1.20)	1.0 (Ref)	1.53 (1.10–2.13)
Model 2	0.88 (0.61–1.25)	1.0 (Ref)	1.54 (1.10–2.14)
Model 3	0.91 (0.64–1.31)	1.0 (Ref)	1.45 (1.03–2.03)
Model 4	0.93 (0.62–1.39)	1.0 (Ref)	1.33 (0.93–1.90)
LiLACS NZ Non-Māori			
OR (95% CI)			
Model 1	1.03 (0.64–1.67)	1.0 (Ref)	1.68 (1.01–2.79)
Model 2	1.41 (0.83–2.41)	1.0 (Ref)	1.95 (1.15–3.33)
Model 3	1.42 (0.83–2.43)	1.0 (Ref)	1.89 (1.11–3.23)
Model 4	1.45 (0.82–2.57)	1.0 (Ref)	1.88 (1.06–3.32)
LiLACS NZ Māori			
OR (95% CI)			
Model 1	0.86 (0.44–1.70)	1.0 (Ref)	0.76 (0.39–1.49)
Model 2	0.86 (0.39–1.89)	1.0 (Ref)	0.83 (0.39–1.76)
Model 3	0.83 (0.37–1.85)	1.0 (Ref)	0.74 (0.34–1.61)
Model 4	0.77 (0.32–1.87)	1.0 (Ref)	0.81 (0.36–1.82)

Model 1 included number of eating occasions and sex; model 2 was also adjusted for living alone, education, deprivation, smoking, and alcohol; model 3 was further adjusted for energy intake; model 4 was further adjusted for disease count, disability, and swallowing problems.

**Table 3 nutrients-13-02574-t003:** Number of eating occasions providing protein according to physical activity level in community-dwelling older adults from the Netherlands (DNFCS), Finland (FINDIET), and Italy (INRAN-SCAI).

	Physical Activity Level			
	Low	Intermediate	High	*p*-Value
DNFCS * (*n* = 1476 days)				
Number of eating occasions				
Model 1	5.4 (0.10)	5.6 (0.05)	5.6 (0.02)	0.057
Model 2	5.4 (0.10)	5.5 (0.06)	5.6 (0.03)	0.070
Model 3	5.4 (0.10)	5.6 (0.06)	5.6 (0.03)	0.16
Exclusion special days	5.4 (0.11)	5.6 (0.05)	5.6 (0.03)	0.22
Exclusion special diets	5.4 (0.12	5.6 (0.06)	5.6 (0.03)	0.25
FINDIET *** (*n* = 1734 days)				
Number of eating occasions				
Model 1	4.4 (0.06)	4.5 (0.03)	4.6 (0.05)	0.0081
Model 2	4.3 (0.07)	4.4 (0.05)	4.5 (0.07)	0.032
Model 3	4.3 (0.07)	4.4 (0.05)	4.5 (0.07)	0.046
Exclusion special days	4.3 (0.07)	4.5 (0.05)	4.5 (0.07)	0.031
Exclusion special diets	N/A	N/A	N/A	N/A
INRAN-SCAI *** (*n* = 1530 days)				
Number of eating occasions				
Model 1	3.8 (0.11)	3.8 (0.06)	4.1 (0.06)	0.0078
Model 2	3.8 (0.15)	3.9 (0.12)	4.1 (0.12)	0.012
Model 3	3.8 (0.14)	3.9 (0.12)	4.1 (0.12)	0.021
Exclusion special days	3.8 (0.15)	3.8 (0.12)	4.0 (0.13)	0.062
Exclusion special diets	3.9 (0.15)	3.9 (0.12)	4.1 (0.12)	0.047

Number of eating occasions presented as mean (standard error). * Sample size (*n*) refers to the number of recall days per survey for participants with non-missing physical activity data. Repeated measures ANOVA, including an unstructured covariance matrix to account for within-person correlation, was used to compare differences according to physical activity level. Model 1: adjusted for age and sex. Model 2: adjusted for age, sex, educational level, marital status, household income (except for INRAN-SCAI) and smoking. Model 3: Model 2 additionally adjusted for energy intake. N/A: not available.

**Table 4 nutrients-13-02574-t004:** Reaching protein thresholds according to physical activity level in community-dwelling older adults in the Netherlands (DNFCS), Finland (FINDIET), and Italy (INRAN-SCAI).

	Physical Activity Level			
	Low	Intermediate	High	*p*-Value
DNFCS (*n* = 1476 days)	*n* = 64	*n* = 252	*n* = 1160	
≥2 eating occasions >20 g protein (%)	28.0 (13.7–42.2)	43.3 (35.7–50.9)	49.9 (45.7–54.2)	0.0043
≥2 eating occasions >30 g protein (%)	1.9 (−6.1–9.9)	7.7 (3.5–12.0)	9.8 (7.4–12.2)	0.12
≥2 main meals >20 g protein (%)	20.5 (6.2–34.8)	41.5 (33.9–49.1)	46.0 (41.7–50.3)	0.0018
≥2 main meals >30 g protein (%)	1.6 (−6.2–9.4)	6.0 (1.9–10.2)	9.0 (6.7–11.3)	0.089
FINDIET (*n* = 1734 days)	*n* = 286	*n* = 1088	*n* = 360	
≥2 eating occasions >20 g protein (%)	37.3 (29.4–45.1)	39.0 (33.4–44.6)	38.0 (30.3–45.7)	0.88
≥2 eating occasions >30 g protein (%)	13.6 (8.4–18.9)	11.9 (8.2–15.7)	14.6 (9.5–19.7)	0.47
≥2 main meals >20 g protein (%)	27.2 (19.8–34.6)	28.3 (23.1–33.6)	27.6 (20.4–34.8)	0.94
≥2 main meals >30 g protein (%)	9.2 (4.60–13.8)	8.4 (5.2–11.7)	8.5 (4.1–13.0)	0.95
INRAN-SCAI (*n* = 1530 days)	*n* = 201	*n* = 642	*n* = 687	
≥2 eating occasions >20 g protein (%)	58.2 (47.3–69.1)	56.7 (47.7–65.7)	62.0 (52.8–71.2)	0.24
≥2 eating occasions >30 g protein (%)	28.5 (18.1–38.9)	30.1 (21.5–38.7)	31.8 (21.5–38.7)	0.73
≥2 main meals >20 g protein (%)	57.3 (46.3–68.3)	56.6 (47.5–65.7)	61.9 (52.6–71.2)	0.23
≥2 main meals >30 g protein	28.5 (18.0–38.9)	29.9 (21.3–38.6)	31.6 (22.8–40.4)	0.74

Number of meals reaching protein thresholds presented as % (95%-CI). Adjusted for age, sex, educational level, marital status, household income, and smoking; *n* is number of recall days per survey for participants with non-missing physical activity data.

## Data Availability

Newcastle 85+ data available on request from https://research.ncl.ac.uk/85plus/datarequests/ (accessed on 13 June 2021). LiLACS NZ data available on request contacting the principal investigator of the study Professor Ngaire Kerse. Data of DNFCS available on request from https://www.wateetnederland.nl/publicaties-en-datasets/datasets (accessed on 13 June 2021). FINDIET data available on request, see more information: https://thl.fi/en/web/thlfi-en/statistics/information-for-researchers (accessed on 13 June 2021). INRAN-SCAI survey data are included in the article/Appendix A, further inquiries can be directed to co-author Stefania Sette.

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
