# Peer review of "Association of Daily Physical Activity and Sedentary Behaviour with Protein Intake Patterns in Older Adults: A Multi-Study Analysis across Five Countries"

_nutrients, 2021, doi:10.3390/nu13082574_

Round 1

Reviewer 1 Report

Comments to the Authors of manuscript number: nutrients-1281268 entitled “Association of daily physical activity and sedentary behaviour with protein intake patterns in older adults: a multi-study analysis across five countries”.

Authors present the study involved elder people from five countries. I would like to congratulate for Authors for their effort and accuracy. It is very interesting study described in many details performed in very large number of people for a long time. The performance of any study on humans is difficult due to the fact that it is difficult to keep self-discipline and complete questionnaire for so long time, especially in elder age.

  1. L9 – nowadays there is rather it described as osteosarcopenia.
  2. L 30 – is seems that physical activity relates to carbohydrate intake as an energy source more that protein. Please add the reference
  3. L 43 – it worth add here the average age about which Authors think
  4. L 68 – it seems that only this line include information about excluded criteria. What about other diseases e.g. related with gastrointestinal tract, metabolic ect.. Were participants asked about illness other that terminal? Many of them can influence not only physical activity but also food intake. Further, older people very frequently suffer from limited mobility of musculoskeletal system. Some people suffer from food intake disorder.
  5. In general it is interesting if participants were divided according to the gender. The demand and food intake is different in females and males, irrespective of the age.
  6. Author presented the recommendation of daily protein intake. Is it for both females and males? Is it dependent on age?

Author Response

We would like to thank the reviewers for their positive feedback and their suggestions to improve the manuscript.

Authors present the study involved elder people from five countries. I would like to congratulate for Authors for their effort and accuracy. It is very interesting study described in many details performed in very large number of people for a long time. The performance of any study on humans is difficult due to the fact that it is difficult to keep self-discipline and complete questionnaire for so long time, especially in elder age.

  1. L9 – nowadays there is rather it described as osteosarcopenia.

Response 1: Osteosarcopenia has been defined as the presence of sarcopenia and osteopenia or osteoporosis (Hassan & Duque, 2017). Cited work from our group and others has focused on sarcopenia rather than its subtypes. Although evidence supports the association of protein intake and bone health, studies examining the association between low protein intake and risk of osteosarcopenia specifically are currently rather limited despite the increasing interest in the condition. We therefore prefer to use the wider term of sarcopaenia. 

  1. L 30 – is seems that physical activity relates to carbohydrate intake as an energy source more that protein. Please add the reference

Response 2: The focus of our paper is on the relationship between physical activity and protein intake patterns. We would prefer not to go into carbohydrate intake and have therefore removed this sentence.

  1. L 43 – it worth add here the average age about which Authors think

Response 3: Thank you, we have added this information to line 43.

  1. L 68 – it seems that only this line include information about excluded criteria. What about other diseases e.g. related with gastrointestinal tract, metabolic etc. Were participants asked about illness other that terminal? Many of them can influence not only physical activity but also food intake. Further, older people very frequently suffer from limited mobility of musculoskeletal system. Some people suffer from food intake disorder.

Response 4: Given the number of studies included in the analysis, we included summaries of the main characteristics of each cohort and survey. However, readers can find detailed information on recruitment strategies and data collection for each study within the references provided in the Methods, section 2.1.

Physical and oral health factors (including common chronic diseases, disability and swallowing problems) were only available in the Newcastle 85+ Study and LiLACS NZ and are described under section 2.6 Socioeconomic, Lifestyle, and Health Factors (lines 237-262). Models were also adjusted for factors such as disability, and their descriptive statistics for Newcastle 85+ Study and LiLACS NZ are shown in Table S1 in the Supplementary material of the manuscript. Information around food intake disorder was not available but in the surveys two sensitivity analyses were undertaken, one excluding special days, and another excluding day on which special diets were consumed (section 2.7).

  1. In general it is interesting if participants were divided according to the gender. The demand and food intake is different in females and males, irrespective of the age.

Response 5: Though we agree with this comment, the two cohort studies would have insufficient power to perform analyses separately by gender. We adjusted models for gender along with other variables and also performed sensitivity analyses using gender-specific cut-points for physical activity in the cohort studies. However, we acknowledge this as a potential limitation and have added this point to our discussion section (lines 563-571).

  1. Author presented the recommendation of daily protein intake. Is it for both females and males? Is it dependent on age?

Response 6: We used the European Food Safety Authority population reference intakes for protein set for adults (0.8 g per kg of body weight per day) which include older adults and are the same for men and women (reference #35).

Reviewer 2 Report

This manuscript and analyses are a novel investigation into the protein consumption of older adults globally and certainly add to the literature in this emerging field. It is well written and the statistical methodology is well supported. However, I do have a few minor questions/clarifications: 

p.5 lines 178-213 - The authors state that they categorized participants into physical activity scores of low, intermediate, and high (and admit there is challenges to harmonization across studies). This is fine, but they do not give any estimate of hours/mins/intensity of how the tertiles in the current analysis were determined, which leads me to wonder if “low, intermediate, and high” are truly comparable between studies. Please add an explanation/definition of “low, intermediate, and high” to the opening paragraph of this section. This is particularly important because the analysis required a sex variable.

p. 6 lines 214-225 The authors outline how sedentary behavior was collected in FINDIET and INRAN-SCAL, but don’t define how the current analysis used this data. Was it combined in a similar textile fashion to physical activity? Please define.

p. 6 lines 235-240 Is there a citation on these BMI categories? Are these WHO cutoffs? Please add a reference here, or outline why these cutoffs were chosen.

Table 1 There are some p-values that are <0.05 (i.e. Newcastle 85+) that do not have indications as to which categories are significantly different, but others that do (i.e. FINDIET). If there are no indications of difference, what is the p value <0.05 indicating? Also, please add the statistical test used in the footnote.

p 15 line 455-469 MAJOR POINT: The authors demonstrate that more physically active older adults have a greater total food consumption than inactive older adults, and thus, are more likely to meet their protein requirements. The authors have defined “meeting requirements” as being >08.g/kg/day. However, previous studies have demonstrated that with increased physical activity comes increased protein requirements of greater than 1.2g/kg/day (Bauer et al, 2013; https://pubmed.ncbi.nlm.nih.gov/23867520/). Based upon the current analysis, only active participants in the LiLACS study (non maori) are meeting this recommendation (with INRAN-SCAI) coming close. As such, is a cutoff of >0.8g/kg/day of protein appropriate for this high activity group? The authors should likely address this in the discussion, rather than coming to the conclusion that simply a higher level of protein for active people is sufficient for “anabolic effects” (line 466).

Author Response

We would like to thank the reviewers for their positive feedback and their suggestions to improve the manuscript.

This manuscript and analyses are a novel investigation into the protein consumption of older adults globally and certainly add to the literature in this emerging field. It is well written and the statistical methodology is well supported. However, I do have a few minor questions/clarifications:  

  1. 5 lines 178-213 - The authors state that they categorized participants into physical activity scores of low, intermediate, and high (and admit there is challenges to harmonization across studies). This is fine, but they do not give any estimate of hours/mins/intensity of how the tertiles in the current analysis were determined, which leads me to wonder if “low, intermediate, and high” are truly comparable between studies. Please add an explanation/definition of “low, intermediate, and high” to the opening paragraph of this section. This is particularly important because the analysis required a sex variable.

Response 1: We have added this information for Newcastle 85+ and LiLACS NZ in section 2.4.  

  1. 6 lines 214-225 The authors outline how sedentary behavior was collected in FINDIET and INRAN-SCAL, but don’t define how the current analysis used this data. Was it combined in a similar textile fashion to physical activity? Please define.

Response 2: We have added this information in section 2.7 (line 302). 

  1. 6 lines 235-240 Is there a citation on these BMI categories? Are these WHO cutoffs? Please add a reference here, or outline why these cutoffs were chosen.

Response 3: Reference had been added. 

  1. Table 1 There are some p-values that are <0.05 (i.e. Newcastle 85+) that do not have indications as to which categories are significantly different, but others that do (i.e. FINDIET). If there are no indications of difference, what is the p value <0.05 indicating? Also, please add the statistical test used in the footnote.

Response 4: Information added to Table 1 footnote. p<0.05 indicates there are significant differences between physical activity groups. The two cohort studies have low power for testing differences between physical activity tertiles. We have therefore removed the indications for the surveys for consistency within Table 1 and across the other tables.

  1. p 15 line 455-469 MAJOR POINT: The authors demonstrate that more physically active older adults have a greater total food consumption than inactive older adults, and thus, are more likely to meet their protein requirements. The authors have defined “meeting requirements” as being >08.g/kg/day. However, previous studies have demonstrated that with increased physical activity comes increased protein requirements of greater than 1.2g/kg/day (Bauer et al, 2013; https://pubmed.ncbi.nlm.nih.gov/23867520/). Based upon the current analysis, only active participants in the LiLACS study (non maori) are meeting this recommendation (with INRAN-SCAI) coming close. As such, is a cutoff of >0.8g/kg/day of protein appropriate for this high activity group? The authors should likely address this in the discussion, rather than coming to the conclusion that simply a higher level of protein for active people is sufficient for “anabolic effects” (line 466).

Response 5: An important point that we have now included in the discussion (lines 493-503).